POINT OF VIEW

# Competency-based assessment for the training of PhD students and early-career scientists

**Abstract** The training of PhD students and early-career scientists is largely an apprenticeship in which the trainee associates with an expert to become an independent scientist. But when is a PhD student ready to graduate, a postdoctoral scholar ready for an independent position, or an early-career scientist ready for advanced responsibilities? Research training by apprenticeship does not uniformly include a framework to assess if the trainee is equipped with the complex knowledge, skills and attitudes required to be a successful scientist in the 21st century. To address this problem, we propose competency-based assessment throughout the continuum of training to evaluate more objectively the development of PhD students and early-career scientists.
DOI: https://doi.org/10.7554/eLife.34801.001

**MICHAEL F VERDERAME[†\*], VICTORIA H FREEDMAN[†], LISA M KOZLOWSKI[†] AND WAYNE T MCCORMACK[†]**

**\*For correspondence:** mxv8@psu.edu

[†]These authors contributed equally to this work

**Competing interests:** The authors declare that no competing interests exist.

The quality of formal training assessment received by PhD students and early-career scientists (a label that covers recent PhD graduates in a variety of positions, including postdoctoral trainees and research scientists in entry-level positions) is highly variable, and depends on a number of factors: the trainee's supervisor or research adviser; the institution and/or graduate program; and the organization or agency funding the trainee. The European approach, for example, relies more on one final summative assessment (that is, a high stakes evaluation at the conclusion of training, e.g. the dissertation and defense), whereas US doctoral programs rely more on multiple formative assessments (regular formal and informal assessments to evaluate and provide feedback about performance) before the final dissertation defense (*Barnett et al., 2017*). Funding agencies in the US such as the National Science Foundation (NSF) and the National Institutes of Health (NIH) have recently increased expectations for formal training plans for individuals supported by individual or institutional training grants (*NIH, 2012*); but these agencies support only a small fraction of PhD trainees via these funding

mechanisms. This variation in the quality and substance of training assessment for PhD students and early-career scientists (*Maki and Borkowski, 2006*) underscores the need for an improved approach to such assessment.

The value of bringing more definition and structure to the training environment has been recognized by professional organizations such as the National Postdoctoral Association, the American Physiological Society/Association of Chairs of Departments of Physiology, and some educational institutions and individual training programs. In addition, a recent NIH Funding Opportunity Announcement places increased emphasis on the development of both research and career skills, with a specific charge that "Funded programs are expected to provide evidence of accomplishing the training objectives". Lists of competencies and skills provide guidelines for training experiences but they are rarely integrated into training assessment plans.

Based on our experience as graduate and postdoctoral program leaders, we recognized the need both to identify core competencies and to develop a process to assess these competencies. To minimize potential confirmation

bias we deliberately chose not to begin this project with a detailed comparison of previously described competencies. Each author independently developed a list of competencies based on individual experiences. Initial lists were wide-ranging, and included traditional fundamental research skills (e.g., critical thinking skills, computational and quantitative skills), skills needed for different career pathways, (e.g., teaching skills), and business and management skills (e.g., entrepreneurial skills such as the ability to develop a business or marketing plan). Although we recognize that many of the competencies we initially defined are important in specific careers, from the combined list we defined 10 core competencies essential for every PhD scientist regardless of discipline or career pathway (*Table 1*).

## Core competencies and subcompetencies

**Broad Conceptual Knowledge of a Scientific Discipline** refers to the ability to engage in productive discussion and collaboration with colleagues across a discipline (such as biology, chemistry, or physics).

**Deep Knowledge of a Specific Field** encompasses the historical context, current state of the art, and relevant experimental approaches for a specific field, such as immunology or nanotechnology.

**Critical Thinking Skills** focuses on elements of the scientific method, such as designing experiments and interpreting data.

**Experimental Skills** includes identifying appropriate experimental protocols, designing and executing protocols, troubleshooting, lab safety, and data management.

**Computational Skills** encompasses relevant statistical analysis methods and informatics literacy.

**Collaboration and Team Science Skills** includes openness to collaboration, self- and disciplinary awareness, and the ability to integrate information across disciplines.

**Responsible Conduct of Research (RCR) and Ethics** includes knowledge about and adherence to RCR principles, ethical decision making, moral courage, and integrity.

**Communication Skills** includes oral and written communication skills as well as communication with different stakeholders.

**Leadership and Management Skills** includes the ability to formulate a research vision, manage group dynamics and communication, organize and plan, make decisions, solve problems, and manage conflicts.

**Survival Skills** includes a variety of personal characteristics that sustain science careers, such as motivation, perseverance, and adaptability, as well as participating in professional development activities and networking skills.

Because each core competency is multi-faceted, we defined subcompetencies. For example, we identified four subcompetencies of Critical Thinking Skills: (A) Recognize important questions; (B) Design a single experiment (answer questions, controls, etc.); (C) Interpret data; and (D) Design a research program. Each core competency has between two to seven subcompetencies, resulting in a total of 44 subcompetencies (*Table 1—source data 1*: Core Competencies Assessment Rubric).

**Table 1.** Ten Core Competencies for the PhD Scientist.

| |
|---|
| 1. Broad Conceptual Knowledge of a Scientific Discipline |
| 2. Deep Knowledge of a Specific Field |
| 3. Critical Thinking Skills |
| 4. Experimental Skills |
| 5. Computational Skills |
| 6. Collaboration and Team Science Skills |
| 7. Responsible Conduct of Research and Ethics |
| 8. Communication Skills |
| 9. Leadership Skills |
| 10. Survival Skills |

DOI: https://doi.org/10.7554/eLife.34801.002

The following source data available for Table 1:
**Source data 1.** Core Competencies Assessment Rubric.
DOI: https://doi.org/10.7554/eLife.34801.005

## Assessment milestones

Individual competencies could be assessed using a Likert-type scale (*Likert, 1932*), but such ratings can be very subjective (e.g., "poor" to "excellent", or "never" to "always") if they lack specific descriptive anchors. To maximize the usefulness of a competency-based assessment rubric for PhD student and early-career scientist training in any discipline, we instead defined observable behaviors corresponding to the core competencies that reflect the development of knowledge, skills and attitudes throughout the timeline of training.

We used the "Milestones" framework described by the Accreditation Council for Graduate Medical Education: "Simply defined, a milestone is a significant point in development. For accreditation purposes, the Milestones are competency-based developmental outcomes (e.g., knowledge, skills, attitudes, and performance) that can be demonstrated progressively by residents and fellows from the beginning of their education through graduation to the unsupervised practice of their specialties."

Our overall approach to developing milestones was guided by the Dreyfus and Dreyfus model describing five levels of skill acquisition over time: novice, advanced beginner, competent, proficient and expert (*Dreyfus and Dreyfus, 1986*). As trainees progress through competent to proficient to expert, their perspective matures, their decision making becomes more analytical, and they become fully engaged in the scientific process (*Dreyfus, 2004*). These levels are easily mapped to the continuum of PhD scientist training: beginning PhD student as *novice*, advanced PhD student as *advanced beginner*, PhD graduate as *competent*, early-career scientist (that includes postdoctoral trainees) as *proficient*, and science professional as *expert* (see *Table 2*).

We therefore defined observable behaviors and outcomes for each subcompetency that would allow a qualified observer, such as a research adviser or job supervisor, to determine if a PhD student or early-career scientist had reached the milestone for their stage of training (*Table 1—source data 1*: Core Competencies Assessment Rubric). A sample for the Critical Thinking Skills core competency is shown in *Table 3*.

## Recommendations for use

We suggest that such a competency-based assessment be used to guide periodic feedback between PhD students or early-career scientists and their mentors or supervisors. It is not meant to be a checklist. Rather than assessing all 44 subcompetencies at the same time, we recommend that subsets of related competencies (e. g., "Broad Conceptual Knowledge of a Scientific Discipline" and "Deep Knowledge of a Specific Field") be considered during any given evaluation period (e.g., month or quarter). Assessors should read across the observable behaviors for each subcompetency from left to right, and score the subcompetency based on the last observable behavior they believe is consistently demonstrated by the person being assessed. Self-assessment and mentor or supervisor ratings may be compared to identify areas of strength and areas that need improvement. Discordant ratings between self-assessment and mentor or supervisor assessment provide opportunities for conversations about areas in which a trainee may be overconfident and need improvement, and areas of strength which the trainee may not recognize and may be less than confident about.

The competencies and accompanying milestones can also be used in a number of other critically important ways. Combined with curricular mapping and program enhancement plans, the competencies and milestones provide a framework for developing program learning objectives and outcomes assessments now commonly required by educational accrediting agencies. Furthermore, setting explicit expectations

**Table 2.** PhD scientist training stages mapped to Dreyfus and Dreyfus levels of skill acquisition. Early-career scientists include researchers undertaking postdoctoral training as well as those in science positions in career pathways that involve other kinds of advanced training, e.g., on-the-job training or certification.

| Dreyfus & Dreyfus | Novice | Advanced beginner | Competent | Proficient | Expert |
|---|---|---|---|---|---|
| | Rule-based behavior, limited, inflexible | Incorporates aspects of the situation | Acts consciously from long-term goals and plans | Sees situation as a whole and acts from personal conviction | Has intuitive understanding of situations, zooms in on central aspects |
| PhD Scientist Training Stages | Beginning PhD Student | Advanced PhD Student | PhD Graduate | Early-Career Scientist | Science Professional |

DOI: https://doi.org/10.7554/eLife.34801.003

**Table 3.** Sample milestones for one of the subcompetencies within Critical Thinking Skills. Verbs in bold font indicate observable behaviors representing each stage of skill acquisition.

| CRITICAL THINKING SKILLS | MILESTONES | | | | |
| --- | --- | --- | --- | --- | --- |
| | Beginning PhD Student | Advanced PhD Student | PhD Graduate | Early-Career Scientist | Science Professional |
| B. Design a single experiment (answer questions, controls, *etc.*) | **Follow** experimental protocols, **seek** help as needed, **describe** critical role of controls | **Plan** experimental protocol; **include** relevant controls; **choose** appropriate methods; **troubleshoot** experimental problems | **Design and execute** hypothesis-based experiments independently; **evaluate** protocols of others; **predict** range of experimental outcomes | Consistently **design and execute** experiments with appropriate controls; **assess** next steps; **critique** experiments of others | **Teach** experimental design; **guide** others doing experiments |

DOI: https://doi.org/10.7554/eLife.34801.004

for research training may enhance the ability of institutions to recruit outstanding PhD students or postdoctoral scholars. Finally, funding agencies focused on the individual development of the trainee may use these competencies and assessments as guidelines for effective training programs.

## Why should PhD training incorporate a competency-based approach?

Some training programs include formal assessments utilizing markers and standards defined by third parties. Medical students, for example, are expected to meet educational and professional objectives defined by national medical associations and societies.

By contrast, the requirements for completing the PhD are much less clear, defined by the "mastery of specific knowledge and skills" (*Sullivan, 1995*) as assessed by research advisers. The core of the science PhD remains the completion of an original research project, culminating in a dissertation and an oral defense (*Barnett et al., 2017*). PhD students are also generally expected to pass courses and master research skills that are often discipline-specific and not well delineated. Whereas regional accrediting bodies in the US require graduate institutions to have programmatic learning objectives and assessment plans, they do not specify standards for the PhD. Also, there are few – if any – formal requirements and no accrediting bodies for early-career scientist training.

We can and should do better. Our PhD students, postdoctoral scholars, early-career scientists and their supervisors deserve both a more clearly defined set of educational objectives and an approach to assess the completion of these objectives to maximize the potential for future success. A competency-based approach fits well with traditional PhD scientist training, which is not bound by a priori finish dates. It provides a framework to explore systematically and objectively the development of PhD students and early-career scientists, identifying areas of strength as well as areas that need improvement. The assessment rubric can be easily implemented for trainee self-assessment as well as constructive feedback from advisers or supervisors by selecting individual competencies for review at regular intervals. Furthermore, it can be easily extended to include general and specific career and professional training as well.

In its recent report "Graduate STEM education for the 21st Century", *The National Academies of Sciences, Engineering, and Medicine, 2018* briefly outlined core competencies for STEM PhDs. In its formal recommendations specifically for STEM PhD education, the first recommendation is, "Universities should verify that every graduate program that they offer provides for these competencies and that students demonstrate that they have achieved them before receiving their doctoral degrees." This assessment rubric provides one way for universities to verify that students have achieved the core competencies of a science PhD.

We look forward to implementing and testing this new approach for assessing doctoral training, as it provides an important avenue for effective communication and a supportive mentor–mentee relationship. This assessment approach can be used for any science discipline, and it has not escaped our notice that it is adaptable to non-science PhD training as well.

## Acknowledgements

We thank our many colleagues at the Association of American Medical Colleges Graduate Research, Education and Training (GREAT)

Group for helpful discussions, and Drs. Istvan Albert, Joshua Crites, Valerie C Holt, and Rebecca Volpe for their insights about specific core competencies. We also thank Drs. Philip S Clifford, Linda Hyman, Alan Leshner, Ravindra Misra, Erik Snapp, and Margaret R Wallace for critical review of the manuscript.

**Michael F Verderame** is in The Graduate School, Pennsylvania State University, University Park, PA, United States

mxv8@psu.edu

http://orcid.org/0000-0001-7046-0879

**Victoria H Freedman** is in the Graduate Division of Biomedical Sciences, Albert Einstein College of Medicine, Bronx, NY, United States

http://orcid.org/0000-0002-7290-0409

**Lisa M Kozlowski** is at Jefferson College of Biomedical Sciences, Thomas Jefferson University, Philadelphia, PA, United States

**Wayne T McCormack** is in the Office of Biomedical Research Career Development, University of Florida Health Sciences Center, Gainesville, FL, United States

https://orcid.org/0000-0002-2117-8727

*Author contributions:* Michael F Verderame, Victoria H Freedman, Lisa M Kozlowski, Wayne T McCormack, Conceptualization, Writing—original draft, Writing—review and editing

*Competing interests:* The authors declare that no competing interests exist.

## Funding

The authors declare that there was no funding for this work.

**Decision letter and Author response**

Decision letter https://doi.org/10.7554/eLife.34801.007
Author response https://doi.org/10.7554/eLife.34801.008

## Additional files

### Data availability

There are no datasets associated with this work

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
