## [Decision Letter]

Thank you for submitting your article "Competency-Based Assessment for the Training of PhD Scientists" to *eLife*. I have assessed your article in my capacity as Associate Features Editor alongside my colleague Peter Rodgers (the *eLife* Features Editor) and a reviewer who has chosen to remain anonymous.

We all enjoyed reading your article and felt that the framework you have developed provides valuable guidance to PhD training that is currently lacking. Indeed, the reviewer stated that "I would look at introducing such a framework at my institution – this benefits both the student and supervisor".

I would therefore like to invite you to submit a revised version of the manuscript that addresses the following points.

Major revisions

1) There are comments about postdocs that do not always fit the narrative of the article. Therefore, it would be best to remove these to focus upon PhD students. Postdoc training is a lot harder to provide a framework for because it could be for anything between 1–5 years. I appreciate the need for such a postdoc framework but overall feel it is best to focus on the PhD aspects.

2) It is not clear from the article whether any PhD scientists have used your competency-based framework yet. Please could you include a section that discusses:

- How extensively the framework has been used so far

- Any assessments you might have performed of the effectiveness of competency-based training, or feedback you've received from PhD scientists and their advisors who have used the framework

- Any plans you have for implementing the approach more widely, or assessing its effectiveness

3) In paragraph four you state that you did not draw your list of competencies from previous lists, but to avoid confirmation bias developed lists based on your own experiences. Please could you discuss how this method avoided bias – could you not have been biased due to reading existing lists some time previously?

4) Mentioning the NSF and the NIH in the first sentence of the second paragraph makes the article very US focused – moving this sentence to later in this paragraph would make the article less US-centric. The article could also incorporate a deeper overview of other countries and practices. For example, many institutions across Europe require students to publish a number of articles before completion of their PhD, and such a goal does provide a "loose" framework for their development.

5) There are a few potential talking points that could be added to the competencies – there is flexibility as to which of the 10 competencies these points could fit. (1) Supervision of other students – this may develop from undergrad supervision through to MSc and lastly new PhD students. (2) An awareness of Innovation/Commercialisation – such as assay or technique development. (3) Public awareness/accountability – This is probably related to ethics but as most PhDs are public or charity funded then there should be a note relating to engagement with funders to understand their funding/operating mechanisms.

---

## [Author Response]

Major revisions1) There are comments about postdocs that do not always fit the narrative of the article. Therefore, it would be best to remove these to focus upon PhD students. Postdoc training is a lot harder to provide a framework for because it could be for anything between 1–5 years. I appreciate the need for such a postdoc framework but overall feel it is best to focus on the PhD aspects.

We agree that a full framework for postdoctoral scholars would be complex, however, acquisition of skills continues after PhD completion, regardless of the first post-graduate position, or the ultimate career path. Accordingly we have renamed this category “Early Career Scientist”, and added a footnote describing this category.

2) It is not clear from the article whether any PhD scientists have used your competency-based framework yet. Please could you include a section that discusses:- How extensively the framework has been used so far- Any assessments you might have performed of the effectiveness of competency-based training, or feedback you've received from PhD scientists and their advisors who have used the framework- Any plans you have for implementing the approach more widely, or assessing its effectiveness

We have not deployed this framework yet but are anxious to do so precisely because we have not yet assessed it – we strongly feel that deploying it in the absence of at least concomitant testing to assess its validity would be unfair to trainees given the risk that it could be inappropriately used as a summative assessment tool (which is most definitely not our intent).

We have shared it with colleagues at several meetings of the AAMC’s Graduate Research Education and Training (GREAT) professional development group (this group consists of biomedical PhD program leaders, and associate deans of graduate education and postdoctoral training at US medical schools). We have also shared it with FASEB’s policy subcommittee on Training and Career Opportunities for Scientists. It has been presented at the NIH (NIGMS) and at a recent joint AAMC-FASEB conference. We have received extremely positive feedback. Based on this feedback, we are anxious to share it more widely, and feel that publication in *eLife* is an excellent vehicle to accomplish this first step.

From the beginning we have been committed to assessing the effectiveness of this framework. As we have presented it, we have had a number of discussions with variety of organizations (scientific societies, private foundations, educational advocacy groups, and others) that have expressed an interest in funding a multi-institutional assessment project. We anticipate that publication would generate additional interest and move the assessment process forward quickly.

3) In paragraph four you state that you did not draw your list of competencies from previous lists, but to avoid confirmation bias developed lists based on your own experiences. Please could you discuss how this method avoided bias – could you not have been biased due to reading existing lists some time previously?

As graduate education leaders we are immersed in the issues of the day, and are certainly aware of and had previously read some of the existing lists of competencies; indeed our extensive experience across the many facets of graduate education and postdoctoral training is what prompted us to start this project in the first place. Thus, we agree with the reviewers that ‘avoiding’ confirmation bias is too strong. We have revised the text to more accurately reflect our meaning: our goal was to minimize confirmation bias, which we did by not contemporaneously reviewing the lists of which we were aware when we initiated this project. We have revised the text accordingly.

4) Mentioning the NSF and the NIH in the first sentence of the second paragraph makes the article very US focused – moving this sentence to later in this paragraph would make the article less US-centric. The article could also incorporate a deeper overview of other countries and practices. For example, many institutions across Europe require students to publish a number of articles before completion of their PhD, and such a goal does provide a "loose" framework for their development.

We have moved the sentences discussing US practices to later in the paragraph. A recent article by Barnett et all (FEBS Open Bio. 7: 1444 (2017)) discussed the lack of formative assessments in the predominant European PhD model in comparison to regular such assessments in the US. While is it true that the US system has regular formative and summative assessments throughout a student’s PhD program (regular dissertation committee meetings, and one or more benchmark examinations during the student’s program), we would argue that the US system is a ‘loose’ framework, to the detriment of our students – hence the importance of this work. We have incorporated this point into the text early in paragraph two.

5) There are a few potential talking points that could be added to the competencies – there is flexibility as to which of the 10 competencies these points could fit. (1) Supervision of other students – this may develop from undergrad supervision through to MSc and lastly new PhD students. (2) An awareness of Innovation/Commercialisation – such as assay or technique development. (3) Public awareness/accountability – This is probably related to ethics but as most PhDs are public or charity funded then there should be a note relating to engagement with funders to understand their funding/operating mechanisms.

1) Supervision of others is already built into many of the assessments; we list several here (out of more than 20 milestones):

a) Competency 2 – Deep Knowledge: “Educate others”, “Train others” (at the Early Career Scientist stage)

b) Competency 3 – Critical thinking skills: “Evaluate protocols of others” (at the PhD Graduate Stage”, “Critique experiments of others (at the Early Career Scientist stage)

c) Competency 4 – Experimental skills: “Assist others” (at the Advanced PhD Student stage), “Help Others” (at the PhD Graduate Stage)

2) “Innovation… such as assay or technique development” is already incorporated into Sub-Competency 2B – Design and execute experimental protocols: “Build a new protocol” (at the Early Career Scientist stage). We determined that “Commercialisation” [sic] and other business and entrepreneurship skills are career specific skills that, while certainly important in some careers, are not a “core” PhD competency. We have slightly elaborated the text to bring this point out.

3) As we understand this comment “Public awareness” falls under Sub-competency 8-F “Communication with the public”. We strongly believe that “Public… accountability” is infused in every aspect of Competency 7 -Responsible Conduct of Research and Research Ethics”.